# Emotion-Focused Mobile App for Promoting Self-Compassion, Self-Protection, and Self-Criticism

**DOI:** 10.3390/ijerph192113759

**Published:** 2022-10-22

**Authors:** Júlia Halamová, Jakub Mihaľo, Lukáš Bakoš

**Affiliations:** Institute of Applied Psychology, Faculty of Social and Economic Sciences, Comenius University in Bratislava, Mlynské luhy 4, 821 05 Bratislava, Slovakia

**Keywords:** self-compassion, self-criticism, self-protection, emotion-focused therapy, mobile app, Mhealth, MHapps

## Abstract

The COVID-19 pandemic has changed our daily lives and restricted access to traditional psychological interventions. Hence there is an immediate and growing demand for accessible and scalable mental health solutions. Emotion-focused training for self-compassion and self-protection was developed and distributed using mobile phone technologies, and its effectiveness was tested. The available research sample consisted of 97 participants with a mean age of 26.06 years and a standard deviation of 10.53. Participants using the mobile app underwent a 14-day program aimed at reducing self-criticism while increasing self-compassion and self-protection. Pre- and post-measurements were collected. The results showed a statistically significant medium effect on self-compassion, self-criticism, and self-protection performance and a significant small effect on self-protection distress. The finding that a 14-day mobile app was able to foster well-being in the form of self-compassion, self-protection, and self-criticism is promising. It indicates the potential for individuals to obtain help through the use of remote tools such as MHapps for a fraction of the usual cost, at their own pace, and without other restrictions.

## 1. Introduction

Blatt and Zuroff [1] defined self-criticism as constant and harsh self-scrutiny accompanied by feelings of unworthiness, shame, inferiority, failure, and guilt [1]. Self-compassion was described by Strauss et al. [2] as the ability to recognize one’s suffering and the universality of human suffering, being able to feel emotional resonance with the suffering, tolerate unpleasant feelings of suffering, and having the motivation to relieve one’s suffering. Both self-compassion and self-criticism may play a crucial role in individuals’ attempts to cope with the consequences of the global pandemic, as these play an important role in well-being and pathology [3,4]. Similarly, a metanalytical study by Kirby et al. [5] reported that self-compassion-based interventions reduce symptoms of anxiety, depression, and stress and increase well-being, mindfulness, and self-compassion. Furthermore, self-compassion protects against feelings of anxiety when facing elevated stress [6] and might be one of the main factors that helps individuals to deal with traumatic stress, which could lead to other psychological issues such as depression, panic disorders, and stress [7]. A meta-analytical study [8] revealed that individuals with healthy levels of self-criticism and self-compassion tend to be happier and score higher on well-being dimensions. In contrast, elevated levels of self-criticism and lower levels of self-compassion could be linked to suicidal ideations [9], perfectionism [10], shame [11], social anxiety, post-traumatic stress disorder, and personality disorders, such as borderline personality disorder, schizophrenia, or major depressive disorder [12,13].

While the benefits of learning to be more self-compassionate and less self-critical are well known, most training still relies on face-to-face interventions which can be costly and hard to access due to suboptimal therapist availability, entry barriers, and a shortage of time [14]. In our view, more accessible ways are needed to provide reliable, efficient, and inexpensive ways of mitigating the negative effects of low self-compassion and high self-criticism. Adapting psychological interventions to “smartphone apps” (MHapps) represents a great opportunity to overcome cost, time, and availability barriers [14]. Smartphone-delivered psychological interventions offer advantages over face-to-face therapy and other technological solutions such as computer programs or web-based online interventions [15]. Most people take their phones with them everywhere they go, and that creates an opportunity to implement evidence-based psychological interventions via this technological device [16]. User engagement is another benefit of smartphones. Users can record their thoughts and emotions in real time, without having to remember all the details, and they can do it wherever they can find privacy [16].

### 1.1. Emotion-Focused Therapy

Emotion-focused therapy (EFT) is an empirically supported psychotherapeutic method [17]. It has been shown to be effective in reducing self-criticism [17] and increasing self-compassion and self-protection [18]. In face-to-face emotion-focused therapy, a therapist guides the client through the discovery of primary adaptive emotions to formulate adaptive information about the situation and create a solution to their problems [19]. By accessing the client’s primary adaptive emotion, the therapist can tackle two primary adaptive feelings—self-protection (also called protective anger) and self-compassion. 

Self-protection helps us to set boundaries when mistreated by others, and self-compassion is the ability to feel compassion towards oneself when faced with suffering [17,18]. Self-protection is closely related to assertiveness—behaviour in which the individual directly and confidently expresses feelings, thoughts, and opinions [20]. Assertiveness training has been linked to a decrease in anxiety, stress, and depression [21] and to better self-image [22], social communication, and happiness [23]. 

It is essential to work with both self-compassion and self-protection because doing so provides an empirically supported treatment for self-criticism [17,18]. Compassion-based interventions have focused primarily on developing compassion and self-compassion in individuals but have failed to address the need to build assertive reactions that can elicit self-protection [24]. The same applies to self-compassionate interventions delivered through mobile technologies, as will be demonstrated later. Self-compassion and self-criticism are both essential aspects of well-being. In recent years, some researchers [25] have studied self-compassion and self-criticism and developed effective interventions to tackle them. However, they have not dealt with self-protection, which is an essential part of the treatment. In the following paragraphs, we outline some of these interventions. 

Gilberts’ [25] compassion-focused therapy (CFT) teaches individuals to feel compassion towards themselves during everyday hardships. It transforms the self-critical “inner voice” into a more compassionate and understanding inner dialogue. There is empirical support for CFT, and it has been shown to be effective in reducing excessive self-criticism [26]. Cultivating emotional balance (CEB) [27] reduces negative emotional states by training individuals to identify and express emotions towards themselves and other people. Group discussions, presentations, guided meditation, and emotional regulation modules have been developed by integrating empirical research and the eastern philosophy of emotions. Several studies have shown the positive impact of CEB in enhancing self-regulation of emotions, mindfulness, self-care, and self-compassion [27,28]. These interventions are delivered through traditional face-to-face sessions, but in recent years, some research papers have focused on creating and testing the efficiency of MHapps for compassionate interventions.

### 1.2. Current Technologies and Self-Compassionate Interventions

Finlay-Jones et al. [29] investigated the effectiveness of self-guided online self-compassion training among psychology trainees. A significant positive effect was found in self-compassion and happiness along with a decrease in depressiveness, distress, and emotional regulation. However, this intervention was not distributed via a mobile app, but provided in an online web-based environment. The disadvantage of that is accessibility. Mobile apps can be accessed anywhere and at any time, but computer-based interventions are only available on a laptop or computer [14].

Krieger et al. [30] created another online self-compassionate intervention for participants with high-level self-criticism. They reported a significant decrease in inadequacy of self, hatred of self, perceived stress, and fear of self-compassion as well as an increase in self-compassion, mindfulness, and life satisfaction. The results were replicated in a follow-up study. 

Similarly, Eriksson et al. [31] created a web-based intervention aimed at measuring the effectiveness of a mindful self-compassion program on stress and burnout symptoms in a group of psychologists. Significant results were obtained for self-compassion, lower self-coldness (the counterpart to self-compassion), stress, and symptoms of burnout [31]. 

There are, of course, some interventions that leverage mobile applications in combination with self-compassionate interventions. Linardon [32], in his systematic and meta-analytic review of randomized controlled trials, suggests that self-compassion and closely related mindfulness could be fostered through mobile-based psychological interventions. However, most of these interventions are based on principles of mindfulness and do not focus solely on self-compassion or self-protection as recommended in the EFT effectiveness research [33]. 

A recent study by Mak et al. [34] discovered that three different mobile app interventions (mindfulness-based, self-compassionate, and cognitive behavioural psychoeducation) were effective in the long term (3-month follow-up) in improving mental well-being and reducing distress among young adults. 

Halamová et al. [24] created emotion-focused training for self-compassion and self-protection (EFT-SCP), which is the only intervention to integrate self-compassion and self-protection and is intended to help individuals reduce self-criticism and increase self-compassion. The intervention [24] was developed based on findings from EFT research as well as other interventions focusing mainly on compassion—Mindful Self-Compassion [35] and Compassion Mind Training [36]. 

The effectiveness of the EFT-SCP was boosted through self-rating measurements and physiological measurements in the study testing the intervention in its in-person group format by Halamová et al. [25] and in clinical samples and [37]. In the study by Halamová et al. [24], e-mail technology was leveraged to instruct participants about 14 daily tasks. It was found to have a significant effect on increasing self-compassion and self-reassurance and reducing self-criticism. These results were sustained at follow-ups. The main disadvantage of the in-person group format is social stigma [38]. In addition, the main shortcoming of the e-mail distribution method is that the researchers had to manually send e-mails and notifications to participants about the daily tasks. This could be solved by creating a mobile app and automatizing the whole process. 

### 1.3. The Aim of the Study

Therefore, we decided to create emotion-focused training for self-compassion and self-protection (EFT-SCP) distributed through mobile phone technology and offering a cost-effective, empirically supported, and constantly available treatment that has an effect on levels of self-criticism and self-compassion and to subsequently test its effectiveness. Our hypotheses were as follows:Attending a fourteen-day emotion-focused training for self-compassion and self-protection (EFT-SCP) via a mobile app will significantly increase self-compassion in participants (similarly, as its in-person group and online e-mail formats of the EFT-SCP [24,33]).Attending a fourteen-day emotion-focused training for self-compassion and self-protection (EFT-SCP) via a mobile app will significantly decrease self-criticism in participants (similarly, as its in-person group and online e-mail formats of the EFT-SCP, [24,33]).Attending a fourteen-day emotion-focused training for self-compassion and self-protection (EFT-SCP) via a mobile app will significantly increase self-protection (assertiveness) in participants (based on the EFT model of change, [39]).

## 2. Materials and Methods

### 2.1. The Research Sample

Participants were recruited through social media, social networking sites, and other online tools such as forums. Particularly, we have found that members of Facebook groups and forums focused on the topic of health, self-improvement, and development were the most interested in trying our intervention. To advertise and recruit participants, we created a short post describing the application and interventions—length, possible benefits, and contact information in case they had any questions. This post also included the link to access the application for more information https://www.self-growth-institute.com/en/home/ (accessed on 15 October 2022). From there, they could access the Google or Apple formats to be downloaded. 

Our inclusion criteria were the following: the participant must be the minimum legal age (18) and the application had to be accessed primarily through an android or IOS device. There were 97 Slovak-speaking participants in the research study, of whom 24.7% were men and 75.3% were women and 59.8% Android and 40.2% IOS users. Out of the 97 participants, 61.9% were single, 18.6% in relationships, 15.5% married, 3.1% divorced, and 1.0% widowed. The education levels were as follows: less than a high school degree 3.1%, high school degree or equivalent 50.5%, some college but no degree 3.1%, bachelor’s degree 14.4%, graduate degree 27.8%, and PhD degree or higher 1.0%. The mean age was 26.06 years with a standard deviation of 10.53, ranging from 18 to 76 years. 

The total number of registered participants using the app was 210 (53.8% drop-out rate). Of the 113 incomplete interventions, most participants did not finish any of the tasks (40–35.4%), 16 participants dropped out on the second day (14.2%), 11 on the third day (9.7%), 8 on the fourth day (7.1%), 5 on the fifth day (4.4%), 3 on the sixth day (2.7%), 6 on the seventh day (5.3%), 4 on the eighth day (3.5%), 2 on the nineth day (1.8%), 3 on the tenth day (2.7%), 1 on the eleventh day (0.9%), 2 on the twelfth day (1.8%), 3 on the thirteenth day (2.7%), and 3 on the last day (2.7%), and 6 participants finished all the exercises but failed to complete the post-intervention questionnaires (5.3%). 

All participants completed an online informed consent form which included the of the goal of the study, possible benefits, their rights to leave at any point with no consequences, risks of participation, and contact information in case of emergency. We have also included information regarding confidentiality—no identifying information resulting from participation was collected or stored, and they have a right to ask for deletion of this information at any point.

Data were collected in accordance with the ethical standards of the institutional and/or national research committee and in accordance with the 1964 Helsinki Declaration and its later amendments or comparable ethical standards. The study’s protocol was approved by the ethical committee of a related university.

### 2.2. Measurement Instruments

#### 2.2.1. The Forms of Self-Criticizing/Attacking & Self-Reassuring Scale (FSCRS)

The Forms of Self-Criticizing/Attacking & Self-Reassuring Scale (FSCRS) was originally developed and tested by Gilbert et al. [40] and was used to test self-criticism and self-reassurance scores in pretest and post-test measurements. The scale contains a total of 22 items measuring three dimensions: reassured self (RS), inadequate self (IS), and hated self (HS). The total self-criticism score is calculated by summing up the IS score (personal inadequacy dimension) and HS score (desire to hurt self dimension). RS refers to the ability to forgive the self, much like self-compassion. Participants rate items on a Likert scale ranging from 0 (“not at all like me”) to 4 (“extremely like me”). An example item for inadequate self is “I feel beaten down by my self-critical thoughts”, for reassured self “I find it easy to forgive myself”, and for hated self “I have a sense of disgust with myself”. The internal consistency of the total self-criticism score is superb at α = 0.90, and the same goes for the individual dimensions (RS = 0.82, IS = 0.86, and HS = 0.80) [41]. Additionally, the FSCRS demonstrates excellent reliability and validity, even for the Slovak general population sample (IS = 0.85, HS = 0.75, and RS = 0.75) [42].

#### 2.2.2. The Sussex-Oxford Compassion for the Self Scale (SOCS-S)

The SOCS-S measures levels of self-compassion in five dimensions. The original scale was developed by Gu et al. [43] (2020) based on the theoretically and empirically grounded definition of self-compassion consisting of: (1) recognizing suffering, (2) understanding the universality of suffering, (3) feeling for the person suffering, (4) tolerating uncomfortable feelings, and finally (5) motivation to act/acting to alleviate suffering. Participants rate items on a Likert scale ranging from 1 (“not at all true”) to 5 (“always true”). Items include recognizing suffering: “I notice when I’m feeling distressed”, understanding the universality of suffering, for example: “I understand that feeling upset at times is part of human nature”, feeling for the person suffering: “when bad things happen to me, I feel caring towards myself”, tolerating uncomfortable feelings, for example: “I connect with my distress without letting it overwhelm me”, and for acting or being motivated to act to alleviate suffering: “when I’m going through a difficult time, I try to look after myself”. The psychometric analysis revealed excellent internal consistency for both an international population (α = 0.91) [43] and the Slovak general population (α = 0.89) [44].

#### 2.2.3. The Short Version of the Scale for Interpersonal Behaviour (s-SIB)

The purpose of the s-SIB is to measure levels of negative assertions (the person’s ability to stand up for him/herself), positive assertions (ability to give and receive compliments), initiating assertiveness (skill of socializing in daily life), and expression of and dealing with personal limitations (ability to cope with pressure, criticism, and failure). The scale consists of 25 items and measures two different phenomena (distress and performance) on a five-point Likert scale. Distress refers to discomfort in daily situations and is rated by strength (not at all, somewhat, rather, very, and extremely), and performance refers to the frequency of each phenomenon (I never do, I rarely do, I sometimes do, I usually do, I always do). Items include negative assertions (“telling someone that you think he/she treated you unfairly”), expression of and dealing with personal limitations (“admitting that you know little about a particular subject”), initiation assertiveness (“starting a conversation with a stranger”), and positive assertions (“acknowledging a compliment on something you have done”). The results of the analysis show good reliability for all the subscales, ranging from α = 0.79 to α = 0.94 [45].

### 2.3. Research Procedure

The study design was a quasi-experiment, meaning there was no active or passive control group. This is because, after downloading the app, a participant from the control group would have been asked to wait for two weeks or do tasks unrelated to the advertised app, which would be discouraging and would probably lead them to delete the app and not do the intervention at all. As the aim was to recruit people from the general public who were interested in the intervention, self-selection and a quasi-experiment was the study design that best matched our requirements. After participants had joined the intervention, they were asked to complete the initial measures for self-compassion (SOCS-S), self-criticism/self-reassurance (FSCRS), and self-protection/assertiveness (s-SIB). The same scales were used at the end of the study. 

### 2.4. EFT-SCP Intervention

EFT-SCP participants underwent a 14-day program aimed at decreasing self-criticism, while increasing self-compassion and self-protection. The EFT-SCP mobile app was developed from previous EFT-SCP interventions, which exist in online-shortened and group-expanded forms [24], and is tailored for the mobile app, which has technical limitations and assets. The final set of exercises was selected by consensus between the first two authors using the following criteria: exercises must follow the core elements of change in emotion-focused therapy for combating self-criticism by developing self-compassion and self-protection [17,18] and must be appropriate for use in a mobile app. 

At the beginning of each task, there was a short section on psychoeducation so participants could learn about the task. The psychoeducation was delivered in text, picture, audio, or video format. The intervention was delivered in the form of expressive writing to deepen the benefits of the intervention [46,47] with 14 specific tasks for each day. Participants were asked questions in the form of text, and some of the exercises included other forms of delivery such as audio files (mediations) and Likert scales. Some of the tasks included extra material (optional activities), where participants could watch videos or study more about the particular subject.

The following questions were used for the expressive writing reflection following task completion and presented on a separate screen for each: how did you feel during the exercise?, which was related to emotional feedback; what did you realize during the exercise?, which was related to cognition feedback; and what did you take from the exercise for everyday life?, which was linked to behaviour feedback. The average time for each exercise was around 15 min, and participants were asked to write their reflections on the task after each daily task. A maximum of one task was allocated to each participant for a 24-h period to allow time for reflection. This also served to check participant adherence to the training.

#### List of Tasks for EFT-SCP


**Day 0: initial measures.**


Filling out the pre-measurements.

**Day 1: How would you treat a friend** (Halamová 2018 [48], inspired by Gilbert [36], Neff 2017 [49], Rockman 2015 [50]). This task was designed to help participants realize how differently we treat ourselves and our friends during times of adversity. B-y evoking a compassionate approach towards a friend, we help them to turn the compassionate language towards themselves. During the first day of the intervention participants are introduced to the main concepts of self-compassion.

**Day 2: Self-compassionate body scan** (Halamová 2018 [48], inspired by Gilbert 2010 [36], Neff 2017 [49], Rockman 2015 [50]). The self-compassionate body scan exercise is a meditation task aimed at relaxing and slowing down participants away from the rapid pace of daily life. Participants are instructed to mindfully notice and sense each part of their body from the legs, up to the chest, head, and then gradually let the tension leave the body.

**Day 3: How self-critical are you?** (Halamová 2018 [48], inspired by Neff 2004 [51]). This exercise guides participants through their own level of self-criticism and self-judgment and their beliefs about them. Respondents reflect on various questions about appearance, career, relationships, school, parenthood, finance, etc. In the second part, participants are guided towards accepting their inadequacies and feeling compassion toward themselves.

**Day 4: My self-criticism** (Halamová 2018 [48], inspired by Gilbert 2010 [36]). This exercise guides participants towards imagining a person that represents their self-critical part (face, facial expressions, gestures, movements, etc.). Participants then have to formulate specific self-critical messages and afterwards state how they would feel, think, and behave and what they would need after the self-criticism. 

**Day 5: Change your self-critical dialogue** (Halamová 2018 [48], inspired by Neff 2017 [49]). In this exercise, participants are instructed to use their knowledge about self-criticism to recognize and learn to change their self-critical voice in everyday life. By recognizing and changing the inner dialogue to self-compassionate messages, they can live a richer and less stressful life. Participants are guided through examples into reformulating their inner self-critical speech to make it more compassionate.

**Day 6: Expressing protective anger—self-protection** (Halamová 2018 [48], inspired by Halamová 2013 [52]). Participants are instructed to recall an event of adversity, criticism, or shame that was aimed at them and to imagine how a close friend would react or protect that person. In the second step, participants reformulate the same message from their own perspective, from themselves to themselves.

**Day 7**: **Self-protective language** (Halamová 2018 [48]). People often use incomprehensible, unsupportive, resigning, helpless, or submissive language towards each other and others. This exercise teaches them to reformulate their inner voice, so it is more supportive of their needs and wants in everyday situations.

**Day 8**: **Assertive rights** (Halamová 2018 [48], inspired by Smith 1975 [53]). The goal of this exercise is to teach assertiveness in everyday life situations. Participants are presented with various assertive rights statements as well as manipulative misbelief so they can compare them and say how differently they would behave if they allowed themselves to be assertive instead of being manipulated in a specific situation. 

**Day 9: Practise saying NO** (Halamová 2018 [48], inspired by Praško 2007 [54]). In this exercise, participants are instructed to create a simple assertive rejection with no explanation or apology and an “emphatic no” that expresses their own feelings, explanation, empathy towards another, and suggests a compromise.

**Day 10: Expressing compassion towards the self** (Halamová 2018 [48], inspired by Halamová 2013 [52]). task requires the participant to recall a moment when they were self-critical and imagine the same situation happening to a vulnerable child. Participants are then instructed to express compassion towards the child and then do the same to themselves. 

**Day 11: A compassionate letter from a friend** (Halamová 2018 [48], inspired by Gilbert 2010 [36], Neff 2017 [49], Rockman 2015 [50]). In this exercise, participants write about what they do not like about themselves and how this makes them feel. In the second part, participants imagine a friend writing a letter from a compassionate perspective about their flaws and inadequacies.

**Day 12: Self-compassionate mirror** (Halamová 2018 [48], inspired by Petrocchi 2017 [55]). Participants are asked to look in the mirror and into their own eyes at the end of the day and practise being self-compassionate about pleasant or unpleasant events that happened to them during the day. Afterwards, they are asked to write about their experience of the task. 

**Day 13. Self-compassionate language** (Halamová 2018 [48]). People often use nonunderstanding, unkind, cold, or critical inner speech towards themselves. This exercise teaches them to reformulate their inner voice, so it is more compassionate, kind, warm, and understanding of themselves in everyday situations.

**Day 14: Self-compassion and self-protection in daily life—planning and practice** (Halamová 2018 [48], inspired by Germer 2016 [56]). This task involves thinking about ways to be more self-compassionate and self-protective physically, emotionally, spiritually, socially, financially, ecologically, and developmentally in daily life. In the second step, they are asked to incorporate the plan into their everyday life.

Day 15–18: after-intervention measures. 

Filling out the post-measurements. 

### 2.5. Data Analysis

The IBM SPSS statistics program version 27 [57] was used for the data analyses. First, we computed Cronbach’s alpha reliability coefficients for each of the scales and their respective dimensions. Then, we calculated the data distribution using the Shapiro–Wilks normality test. For data with a normal distribution, we used the paired t-test and calculated Cohen’s d as the measure of the effect size. Where the data distribution was violated, we used the Wilcoxon signed-rank test Mann–Whitney U test to calculate the effect size. 

## 3. Results

The reliability coefficients for each of the scales and their subscales and the results of the Shapiro–Wilks normality test are included in Table 1. 

### 3.1. Self-Compassion

The total SOCS-S self-compassion score showed a statistically significant effect of medium strength, *Z* = −4.546; *p* = 0.000; r_m_ = 0.46. Furthermore, we found a statistically significant and medium-size effect for three SOCS-S subscales, namely (FS) feeling for the person suffering, *Z* = −4.548; *p* = 0.000; r_m_ = 0.45, (TS) tolerating uncomfortable feelings, *t* = −4.454; *df* = 96; *p* = 0,000; *d* = 0.64, and (MA) motivation to act/acting to alleviate suffering, *Z* = −4.849; *p* = 0.000; r_m_ = 0.49. There was no statistically significant difference between the pretest and post-test scores reported for the SOCS-S RS recognizing suffering dimension (*Z* = −1.190; *p* = 0.234; r_m_ = 0.24) or for the SOCS-S US understanding the universality of suffering (*Z* = −1.164; *p* = 0.244; r_m_ = 0.24).

### 3.2. Self-Criticism

We found a statistically significant and medium-size effect in the level of self-criticism in the FSCRS IS+HS pretest and post-test measurements, *t* = 4.181; *df* = 96; *p* = 0.000; *d* = 0.60. A statistically significant and medium-size effect was obtained for the FSCRS inadequate self dimension, *t* = 4.474; *df* = 96; *p* = 0.000; *d* = 0.65, and for the FSCRS reassured self dimension, *Z* = −5.405; *p* = 0.000; r_m_ = 0.55. No statistically significant difference was reported between the pretest and post-test scores for the FSCRS hated self subscale (*Z* = −1.535; *p* = 0.125; r_m_ = 0.31).

### 3.3. Self-Protection

We found a statistically significant and small-size effect in the level of self-protection/assertiveness in the pretest and post-test measurements for s-SIB distress, *Z* = −2.099; *p* = 0.036; r_m_ = 0.21, as well as the positive assertions (PA) subscale, *Z* = −2.861; *p* = 0.004; r_m_ = 0.29. We found no statistically significant difference in pretest and post-test measurements for (NA) negative assertions (*t* = 1.185; *df* = 96; *p* = 0.239; *d* = 0.17), (PL) expression of and dealing with personal limitations subscale (*Z* = −1.922; *p* = 0.055; r_m_ = 0.20), or (IA) initiating assertiveness (*Z* = −1.902; *p* = 0.057; r_m_ = 0.19). 

A statistically significant and medium-size effect was found for the total score for s-SIB performance, *t* = −3.906; *df* = 96; *p* = 0.000; *d* = 0.56, and the subscales (PA) positive assertions, *Z* = −3.566; *p* = 0.000; r_m_ = 0.36, and (NA) negative assertions, *t* = −3.658; *df* = 96; *p* = 0,000; *d* = 0.53. There was no statistically significant difference between the pre-test and post-test scores reported for the dimensions (PL) expression of and dealing with personal limitations (*Z* = −1.922; *p* = 0.055; r_m_ = 0.20) and (IA) initiating assertiveness (*t* = 1.470; *df* = 96; *p* = 0.145; *d* = 0.21).

## 4. Discussion

The current study inspected the immediate effects of a mobile app version of the empirically supported 14-day emotion-focused training for self-compassion and self-protection (EFT-SCP) on self-compassion, self-protection, and self-criticism. All the hypotheses were supported by the finding that attending EFT-SCP via a mobile app yielded a significant increase in self-compassion and self-protection and a significant decrease in self-criticism in participants. The results support previous studies of EFT-SCP effectiveness [33,37], although these previous studies used different means of delivery, such as personal groups or online, and slightly different measurements: the physiological indicator HRV and self-compassion were measured using the self-compassion scale from Neff [58]. Our results correspond with previous findings showing that self-compassionate interventions distributed through technologies such as MHapps and web applications raised self-compassion [29,31,32] and decreased self-criticism [30].

High self-compassion and low self-criticism are both essential aspects of well-being. As mentioned in the literature review, lower levels of self-criticism and higher levels of self-compassion are linked to a happier life on well-being dimensions [8] and a decrease in symptoms of depression, anxiety, and stress [6,59]. Conversely, suboptimal levels of self-compassion and high levels of self-criticism may elicit suicidal thoughts [9], unhealthy perfectionism [10], shame [11], and other serious mental issues such as PTSD, major depression, BPO, and schizophrenia [12,13]. Therefore, our EFT-SCP mobile app has the potential to improve well-being and decrease pathology, as the used scales are validated in Slovakia, and there are norms for them [42,45], so we know that the baseline average measurements were within the average scores. In the Slovak general population, at the beginning and after the intervention, they significantly changed in expected directions.

Using previously tested face-to-face interventions [36] and leveraging the current technologies can help address the expense, unavailability of professionals, and stigma of traditional interventions [38]. One of the most prevalent barriers of entry into traditional psychotherapeutic sessions is the stigma [38]. Surprisingly, the stigma attached to mental health issues is more prevalent in younger rather than the elderly population [60]. In our view, merging the massively adopted technology of smartphones and implementing psychological interventions would be an ideal strategy to combat the stigma attached to mental health issues. We hypothesize that MHapps could be considered an entry point to the world of mental health care and would help to destigmatize psychological or psychiatric treatments. 

Furthermore, the COVID-19 pandemic has changed the way we work, travel, entertain, eat, and interact in daily life, and in some cases, it restricted our access to traditional psychological interventions. In this context, we argue that there is an immediate increase in demand for accessible and scalable mental health solutions (e.g., mobile health and electronic health) to promote well-being and prevent mental disorders in the current situation. Therefore, it is promising that the use of a 14-day mobile app fostered well-being in terms of self-compassion, self-protection, and self-criticism. Our results are in line with a recent systematic and meta-analytic review of randomized controlled trials [32]. The present study demonstrates that it is possible to help individuals via remote tools such as MHapps for a fraction of the usual cost, at their own pace, and without other restrictions. 

Moreover, self-protection is an essential part of emotion-focused therapy and provides a plethora of benefits such as lowering anxiety, stress, and depression and enhancing happiness, self-image, and social communication [21,22,23]. The most recent psychological self-compassion interventions distributed online or via mobile apps do not address assertiveness and self-protection [29,30,31,32,34]. As the current EFT effectiveness research indicates, the most effective interventions for combatting self-criticism include both self-compassionate and self-protective elements [33]. Not surprisingly, there were larger effect sizes in the change in self-protection performance than in self-protection distress. That is because it is easier and quicker to change behaviour than it is to change emotions acquired through long-term habits [61]. Including self-protective tasks in the intervention resulted in a partial increase in self-protection performance. We assume that prolonging the EFT-SCP intervention, subsequent long-term practice, or including a larger number of self-protective exercises would yield a bigger effect size on the assertiveness dimensions for both performance and distress. 

### 4.1. Future Research

In the future, it would be beneficial to ascertain how effective this mobile app is in translation. Currently, data on the effectiveness of the English version of EFT-SCP are being collected. 

### 4.2. Implications for Practice

As self-criticism is a transdiagnostic phenomenon related to all kinds of psychopathology [13], it is very promising that a readily available mobile app that does not require direct contact with a health professional may have a preventive and positive effect on self-criticism. In addition, there is hope that it will increase self-compassion and self-protection as well because they are both related to well-being and good mental and physical health [8]. 

### 4.3. Limitations

The main limitation of this study is not having a control group to compare our results with. Hence it is possible that the effect was not due to participation in the mobile app, but to changes over time. The second important limitation is that the follow-up effects of the mobile app were not measured, so we do not know whether the changes are long term or just short term. Lastly, participants were self-selected based on motivation and possession of a mobile or computer to access the mobile app.

## 5. Conclusions

The 14-day mobile app version of the empirically supported emotion-focused training for self-compassion and self-protection (EFT-SCP) significantly increased self-compassion, self-reassurance, self-protection in the form of distress, and performance and significantly decreased self-criticism, with effects evident right after the intervention. The results suggest that interventions similar to this could be used in prevention to reduce the threat of an evolving psychopathology. In addition, the mobile app has the potential to make EFT-SCP available to wide-ranging populations of possible participants without the need for direct contact with a professional. Populations unable to access or too ashamed to contact a health care provider could benefit immensely from this type of intervention delivery. 

## Figures and Tables

**Table 1 ijerph-19-13759-t001:** The results of the Shapiro–Wilks normality test and Cronbach’s alpha for internal consistency.

	Shapiro–Wilkes Normality Test						Cronbach’s α
Scales and Dimensions	Pretest			Post-Test			
	Statistic	Df	Sig.	Statistic	Df	Sig.	
FSCRS IS	0.980	97	0.137	0.984	97	0.271	0.887
FSCRS RS	0.975	97	0.062	0.946	97	0.001	0.840
FSCRS HS	0.913	97	0.000	0.896	97	0.000	0.735
FSCRS IS + HS	0.989	97	0.636	0.974	97	0.051	0.900
SOCS-S	0.967	97	0.015	0.971	97	0.028	0.902
SOCS-S RS	0.863	97	0.000	0.845	97	0.000	0.842
SOCS-S US	0.726	97	0.000	0.599	97	0.000	0.756
SOCS-S FS	0.968	97	0.018	0.973	97	0.041	0.779
SOCS-S TS	0.978	97	0.111	0.981	97	0.160	0.694
SOCS-S MA	0.970	97	0.025	0.945	97	0.001	0.826
s-SIB	0.979	97	0.117	0.972	97	0.034	0.916
s-SIB NA	0.987	97	0.487	0.986	97	0.369	0.699
s-SIB PL	0.958	97	0.004	0.956	97	0.002	0.764
s-SIB IA	0.983	97	0.254	0.976	97	0.073	0.704
s-SIB PA	0.965	97	0.011	0.966	97	0.012	0.726
s-SIB F	0.987	97	0.454	0.977	97	0.092	0.926
s-SIB FNA	0.991	97	0.797	0.988	97	0.533	0.734
s-SIB FPL	0.986	97	0.399	0.960	97	0.005	0.784
s-SIB FIA	0.983	97	0.254	0.976	97	0.073	0.713
s-SIB FPA	0.975	97	0.066	0.971	97	0.033	0.802

## Data Availability

Data are available upon request for researchers who consent to adhering to the ethical regulations for confidential data.

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
