# Peer review of "Emotion-Focused Mobile App for Promoting Self-Compassion, Self-Protection, and Self-Criticism"

_ijerph, 2022, doi:10.3390/ijerph192113759_

Round 1

Reviewer 1 Report

This study focuses on the implementation of a 14-day emotion-focused training through a mobile application, in order to subsequently test its effectiveness on the levels of self-compassion, self-criticism and self-protection. The authors adequately support this intervention through a good theoretical basis and state all relevant aspects of the intervention clearly in the manuscript. I believe that the findings may have important implications for practice, as the authors themselves have been able to highlight. Despite the limitations and the need for further studies, I consider this to be an excellent work and a great scientific contribution.

Introduction

The introduction reflects comprehensively and adequately the previous literature on the subject. The authors have put together a broad theoretical framework that allows for an understanding of the study as a whole, addressing all relevant concepts. I would like to congratulate the authors for this. Just a few small suggestions:

When talking about emotion-focused training, the terms self-protection and protective anger are used interchangeably. Although in parentheses you clarify that they are the same, I think it would be more useful for the proper understanding of the text to use only the term self-protection. The use of the term "protective anger" at some points (e.g., lines 77-78) could be confusing. 

On the other hand, some paragraphs may be too long, making them difficult to understand. Although I find all the information relevant and necessary, sometimes it can be difficult to follow the main idea of the text among all the studies mentioned. Therefore, I suggest splitting some paragraphs, for example, those in lines 63-87 or 103-127.

Materials and methods

The information presented regarding the method is adequate and clearly stated. In particular, congratulations to the authors for the development and excellent presentation of the intervention used. Even so, I propose a few small changes to improve this section.

First, with respect to the participants, I consider it necessary to explain further how the sample was obtained. Participants were reportedly recruited through social networks and online tools such as forums. I think it would be appropriate to explain which social networks or specific tools were used. In addition, information should be provided on how potential participants accessed the links or were informed of the existence of the study.

It would also be useful to explain what information was given to participants before accepting informed consent: purpose of the study, possibility of leaving the study, anonymity, etc.

As for the intervention, it could be considered to include it as a subsection within the instruments or within the procedure. Only in order not to break the classic division of the information into the subsections of participants, measurements, procedure and statistical analysis. This is just a suggestion. In this regard, what I would recommend is that the statistical analyses be a subsection of the materials and methods section, instead of having their own section.

Also, some of the intervention tasks could be further developed. In particular, in the tasks that require instructing the participants (such as the one on day 9 or 10), it could be explained what instructions are given to them to achieve the proposed objectives.

Finally, please check the title of the second paragraph. This journal proposes that the title should be "materials and methods", instead of "methods".

Results

The results are presented in full, showing the findings of each of the analyses performed.

Only, I think it would be better to present the results of the Shapiro-Wilks Normality Test in a table format to make it more visual and organized. In addition, the results of the internal consistency analysis (Cronbach's alpha) could be added to this table and presented together, without the need to divide both between the results section and the annexes.

Discussion

This section is very complete and consistent with the findings obtained, putting them in relation to previous literature and highlighting the major implications of this study, as well as clearly stating the limitations. I would like to congratulate the authors for this.

As suggestions, on lines 414-416 I think you could slightly develop the idea that these types of interventions can help address the stigma of traditional interventions. I think it might suffice to explain why this happens or could happen, and what implications it may have in practice.

In addition, I think it might be useful to explain how self-compassion and self-protection affect physical and mental health. That is, briefly detail which aspects of physical and mental health are affected by these constructs.

Author Response

Thank you for personally reviewing our manuscript and for giving us the opportunity to respond to your comments. We have made the relevant changes as detailed below and would like to resubmit our amended manuscript We hope you find our response detailed and satisfactory. If you have any further questions or comments, please do not hesitate to contact me.

Authors

REVIEWER 1

Comments and Suggestions for Authors

This study focuses on the implementation of a 14-day emotion-focused training through a mobile application, in order to subsequently test its effectiveness on the levels of self-compassion, self-criticism and self-protection. The authors adequately support this intervention through a good theoretical basis and state all relevant aspects of the intervention clearly in the manuscript. I believe that the findings may have important implications for practice, as the authors themselves have been able to highlight. Despite the limitations and the need for further studies, I consider this to be an excellent work and a great scientific contribution.

Thank-you for the positive feedback, much appreciated:)

Introduction

The introduction reflects comprehensively and adequately the previous literature on the subject. The authors have put together a broad theoretical framework that allows for an understanding of the study as a whole, addressing all relevant concepts. I would like to congratulate the authors for this.

Thank you!

Just a few small suggestions:When talking about emotion-focused training, the terms self-protection and protective anger are used interchangeably. Although in parentheses you clarify that they are the same, I think it would be more useful for the proper understanding of the text to use only the term self-protection. The use of the term "protective anger" at some points (e.g., lines 77-78) could be confusing. 

Thanks. We have unified terminology and are using only self-protection to describe this phenomenon.

On the other hand, some paragraphs may be too long, making them difficult to understand. Although I find all the information relevant and necessary, sometimes it can be difficult to follow the main idea of the text among all the studies mentioned. Therefore, I suggest splitting some paragraphs, for example, those in lines 63-87 or 103-127.

Thank-you. We have changed the paragraphs to be shorter and easier to understand.

Materials and methods

The information presented regarding the method is adequate and clearly stated. In particular, congratulations to the authors for the development and excellent presentation of the intervention used.

Thank you.

Even so, I propose a few small changes to improve this section.First, with respect to the participants, I consider it necessary to explain further how the sample was obtained. Participants were reportedly recruited through social networks and online tools such as forums. I think it would be appropriate to explain which social networks or specific tools were used. In addition, information should be provided on how potential participants accessed the links or were informed of the existence of the study.

Thanks a lot. We have added a more detailed and comprehensive explanation of the data gathering process. During our data collection process we have found that members of Facebook groups and forums focused on the topic of health, self-improvement, and development were the most keen to try our intervention. To advertise and recruit participants we have created a short post describing the application and interventions – length, possible benefits, and contact in-formation in case they had any questions. This post also included the link to access the application for more information https://www.self-growth-institute.com/en/home/. From there they could access the Google or Apple where where the app can be downloaded.

It would also be useful to explain what information was given to participants before accepting informed consent: purpose of the study, possibility of leaving the study, anonymity, etc.

Thank you. We have added a more detailed description of the data collection and rights of participants (lines 197-202).

As for the intervention, it could be considered to include it as a subsection within the instruments or within the procedure. Only in order not to break the classic division of the information into the subsections of participants, measurements, procedure and statistical analysis. This is just a suggestion. In this regard, what I would recommend is that the statistical analyses be a subsection of the materials and methods section, instead of having their own section.

Thanks. We have included „data analysis“in the methods section and changed the numbering of the following sections.

Also, some of the intervention tasks could be further developed. In particular, in the tasks that require instructing the participants (such as the one on day 9 or 10), it could be explained what instructions are given to them to achieve the proposed objectives.

Thanks. However, the reason for not including a more detailed description of each task was that we aimed to make our text more concise. We have added more information to lines 268-275 to adress your constructive comment. The previous section which described EFT-SCP intervention describes this proces:

  1. At the beginning of each task there was a short section on psychoeducation so partici-pants could learn about the task
  2. The intervention was delivered in expressive writing to deepen the benefits of the intervention (Pennebaker, 2017; Pennebaker & Beall, 1986) with 14 specific tasks for each day.
  3. The following questions were used for the expressive writing reflection following task completion and presented on a separate screen for each: How did you feel during the exercise?, which was related to emotion feedback; What did you realize during the exercise?, which was related to cognition feedback; and What did you take from the exercise for everyday life?, which was linked to behaviour feedback.
  4. Participants were asked to write their reflections on the task after each daily task.

Finally, please check the title of the second paragraph. This journal proposes that the title should be "materials and methods", instead of "methods".

Thank you. This was corrected.

Results

The results are presented in full, showing the findings of each of the analyses performed.

Only, I think it would be better to present the results of the Shapiro-Wilks Normality Test in a table format to make it more visual and organized. In addition, the results of the internal consistency analysis (Cronbach's alpha) could be added to this table and presented together, without the need to divide both between the results section and the annexes.

Thanks a lot. We have formated both Shapiro-Wilks and Cronbach alpha data into APA format tables and added them to the results section. Therefore, the Appendix is deleted.

Table 1

The results of Shapiro-Wilks Normality and Cronbach’s alpha for internal consistency

                Shapiro-Wilkes Normality Test

Cronbach’s α

Scales and dimensions

Pretest

Postest

Statistic

Df

Sig.

Statistic

Df

Sig.

FSCRS IS

0.980

97

0.137

0.984

97

0.271

0.887

FSCRS RS

0.975

97

0.062

0.946

97

0.001

0.840

FSCRS HS

0.913

97

0.000

0.896

97

0.000

0.735

FSCRS IS+HS

0.989

97

0.636

0.974

97

0.051

0.900

SOCS-S

0.967

97

0.015

0.971

97

0.028

0.902

SOCS-S RS

0.863

97

0.000

0.845

97

0.000

0.842

SOCS-S US

0.726

97

0.000

0.599

97

0.000

0.756

SOCS-S FS

0.968

97

0.018

0.973

97

0.041

0.779

SOCS-S TS

0.978

97

0.111

0.981

97

0.160

0.694

SOCS-S MA

0.970

97

0.025

0.945

97

0.001

0.826

s-SIB

0.979

97

0.117

0.972

97

0.034

0.916

s-SIB NA

0.987

97

0.487

0.986

97

0.369

0.699

s-SIB PL

0.958

97

0.004

0.956

97

0.002

0.764

s-SIB IA

0.983

97

0.254

0.976

97

0.073

0.704

s-SIB PA

0.965

97

0.011

0.966

97

0.012

0.726

s-SIB F

0.987

97

0.454

0.977

97

0.092

0.926

s-SIB FNA

0.991

97

0.797

0.988

97

0.533

0.734

s-SIB FPL

0.986

97

0.399

0.960

97

0.005

0.784

s-SIB FIA

0.983

97

0.254

0.976

97

0.073

0.713

s-SIB FPA

0.975

97

0.066

0.971

97

0.033

0.802

Discussion

This section is very complete and consistent with the findings obtained, putting them in relation to previous literature and highlighting the major implications of this study, as well as clearly stating the limitations. I would like to congratulate the authors for this.

Thank you!

As suggestions, on lines 414-416 I think you could slightly develop the idea that these types of interventions can help address the stigma of traditional interventions. I think it might suffice to explain why this happens or could happen, and what implications it may have in practice.

We appreciate this suggestion a lot. We have added an explanation of the stigma-related issue on lines 434-440.

In addition, I think it might be useful to explain how self-compassion and self-protection affect physical and mental health. That is, briefly detail which aspects of physical and mental health are affected by these constructs.

The impact of self-compassion on mental health is described on lines 418-425. The impact of self-protection is described on lines 71-77. We did not include the impact of these phenomena to physical health as this was not the main goal of the study.

Many studies suggest that mental and physical worlds are connected and affect one another. Eg. Launders, N., Dotsikas, K., Marston, L., Price, G., Osborn, D. P., & Hayes, J. F. (2022). The impact of comorbid severe mental illness and common chronic physical health conditions on hospitalisation: A systematic review and meta-analysis. PLOS ONE, 17(8). https://doi.org/10.1371/journal.pone.0272498

However, this is beyond the scope of our study and that is the reason that only effects on mental health were described.

Reviewer 2 Report

The paper is very interesting. I was curious to read all of it. The authors provide sufficient information on the background and clearly explain their methods and results. Some issues i noticed are listed below:  

The argument of 14-days is not explained in the hypotheses. Why do the authors assume in their hypotheses that ' Attending fourteen-day Emotion-Focused Training' for the expected results is important? Later in the text I could see that this is because of the 14 daily tasks. However, if there are not any explainable reasons on the selection of this duration, it would then this claim should be removed from the hypotheses statements.

  Overall the hypotheses are not well-argumented, based on the theory/citations. The authors could change them towards research questions so their statements can remain generic.   The citation/ study source is missing from the  Short Version of the Scale for Interpersonal Behavior (s-SIB)
  The daily exercises are  clearly described. I would like to see some more information on the procedure and the learning mean. E.g., the exercises were all in a written format? Was there any multimedia training material?
  Minor issues: font size looks different at lines 318-320, the same in other parts e.g. l.249-250, l.387-393

Author Response

Thank you for personally reviewing our manuscript and for giving us the opportunity to respond to your comments. We have made the relevant changes as detailed below and would like to resubmit our amended manuscript We hope you find our response detailed and satisfactory. If you have any further questions or comments, please do not hesitate to contact me.

Authors

REVIEWER 2

The paper is very interesting. I was curious to read all of it. The authors provide sufficient information on the background and clearly explain their methods and results.

Thank you!

Some issues i noticed are listed below:  

The argument of 14-days is not explained in the hypotheses. Why do the authors assume in their hypotheses that ' Attending fourteen-day Emotion-Focused Training' for the expected results is important? Later in the text I could see that this is because of the 14 daily tasks. However, if there are not any explainable reasons on the selection of this duration, it would then this claim should be removed from the hypotheses statements.

Thanks a lot. The rationale behind „14-day intervention would change the measured outcomes“ comes from previous studies researching this intervention. In particular,

Online email format EFT-SCP intervention tested by

Halamová, J., Kanovský, M., Varšová, K., & Kupeli, N. (2018). Randomised controlled trial of the new short-term online emotion focused training for self-compassion and self-protection in a nonclinical sample. Current Psychology, 40(1), 333–343. https://doi.org/10.1007/s12144-018-9933-4

Group in person format EFT-SCP intervention tested by

Halamová, J., Koróniová, J., Kanovský, M., Kénesy Túniyová, M., & Kupeli, N. (2019). Psychological and physiological effects of emotion focused training for self-compassion and self-protection. Research in Psychotherapy: Psychopathology, Process and Outcome, 22(2). https://doi.org/10.4081/ripppo.2019.358

Therefore, we had no reason to assume that delivering these interventions via mobile application would have a different or no effect. The info is added into the manuscript.

Overall the hypotheses are not well-argumented, based on the theory/citations. The authors could change them towards research questions so their statements can remain generic.  

Thank you, great point. We have chosen to create hypotheses instead of research questions due to previous studies researching the same intervention. Citations were added directly following each hypothesis to state this fact more clearly.

The citation/ study source is missing from the  Short Version of the Scale for Interpersonal Behavior (s-SIB)

The daily exercises are  clearly described. I would like to see some more information on the procedure and the learning mean. E.g., the exercises were all in a written format? Was there any multimedia training material?

We appreciate your suggestion. We have used a combination of questions (text), pictures, videos, audio recordings, and Likert scale questionnaires. The video was input via youtube and was accessible directly through the app. Audio files were mostly used for meditation exercises. These were recorded in collaboration with a professional voice actress. Thank you for the suggestion. We have added to the process description (see lines 265-272).

Minor issues: font size looks different at lines 318-320, the same in other parts e.g. l.249-250, l.387-393

Thank you. This was corrected.

Reviewer 3 Report

1-    General Comments:

-This paper addresses the impact of the utilization of a mobile app to deliver a psychological intervention to a normal population

- It is a interesting work with clinical importance in the area of healthcare

-The  methodology chosen is adequate to the aims that were delineated

-The paper is written in a clear language and in a systematic way but sometimes it seems like a work resulting from an academic thesis and these parts should be improved

-I do not agree with the expression used related to the intervention as «a mere 14-day mobile app» it devalues the app

-There are some mistakes relating to letter size along the manuscript, please correct it

a.     Abstract:  

-It is well structured and addresses the key findings with quantitative data and implications of this work to the clinical practice

b.    Keywords

-They are adequate

a.     Introduction:

- The paper has a focused state of art strengthening the need for this investigation

-It was important to include a rationale/appropriate justification for the general methodology chosen

- The authors should better depict, what their study adds to our current knowledge regarding the aspects of the impact of  this intervention compared with other types of psychological interventions, that seemed to be a central dimension of this work

b.     Methods:

-Participants should be more extensively characterized as well as inclusion criteria and exclusion criteria

- Ethical procedures were well detailed (informed consent obtainment, etc)

-Statistical methods are well described and are adequate to study design

-The absence of a control group can be a limitation in the procedures

-There is a  detailed program description would enrich the manuscript.

- It would be interesting to use a flow chart related to participants participation

c) Results:

-Results are well described

-Probably the results from reliability coefficients would be more systematic presented in a table

d) Discussion:

-Data are well discussed but should be more articulated with the state of art

-It could be interesting to explore the association with expressive writing of Pennebaker, it was mentioned but only shortly

-I do not agree with this argument, I think it would be very important to have some baseline evaluations of the target population «Therefore, our EFT-SCP mobile app has the potential to improve well-being and decrease pathology, although we did not measure these important psychological constructs because we did not want to overwhelm participants by requiring them to complete so many questionnaires.»

-However it would be interesting to discuss some implications of the program nature (techniques and theoretical assumptions), comparisons between the different approaches used in an app and face to face

-The relevance for clinical practice is well established

Author Response

Thank you for personally reviewing our manuscript and for giving us the opportunity to respond to your comments. We have made the relevant changes as detailed below and would like to resubmit our amended manuscript We hope you find our response detailed and satisfactory. If you have any further questions or comments, please do not hesitate to contact me.

Authors

REVIEWER 3

1-    General Comments:

 -This paper addresses the impact of the utilization of a mobile app to deliver a psychological intervention to a normal population

- It is a interesting work with clinical importance in the area of healthcare -The  methodology chosen is adequate to the aims that were delineated -The paper is written in a clear language and in a systematic way but sometimes it seems like a work resulting from an academic thesis and these parts should be improved

Thank-you for the positive feedback!

-I do not agree with the expression used related to the intervention as «a mere 14-day mobile app» it devalues the app

Thank you. You are correct and the word “mere” was deleted.

-There are some mistakes relating to letter size along the manuscript, please correct it

This was corrected. Thank you.

  1. Abstract: 

-It is well structured and addresses the key findings with quantitative data and implications of this work to the clinical practice

Thank you.

  1. Keywords 

-They are adequate

  1. Introduction:

 - The paper has a focused state of art strengthening the need for this investigation

-It was important to include a rationale/appropriate justification for the general methodology chosen. The authors should better depict, what their study adds to our current knowledge regarding the aspects of the impact of  this intervention compared with other types of psychological interventions, that seemed to be a central dimension of this work

Thank you. The rationale for this study was to create and test the efficiency of mobile applications on self-compassion, self-criticism, and self-protection. The reasons for this study are described in lines 144-153. We added more explanation to it.

  1. Methods:  

-Participants should be more extensively characterized as well as inclusion criteria and exclusion criteria

Thank you. We have described participants more extensively (used device, education, martial status, and country) and provided inclusion/exclusion criteria.

- Ethical procedures were well detailed (informed consent obtainment, etc).

Thank you. This was added to The research sample section (see lines 185-190).

-Statistical methods are well described and are adequate to study design

Thank you.

-The absence of a control group can be a limitation in the procedures

Absolutely, we agree. We have reflected on this fact in lines 248-254. We have chosen quasi-experiment because after downloading the app a participant from the control group is asked to wait for two weeks or do tasks unrelated to the advertised app, which would be discouraging and would probably lead them to delete the app and not do the intervention at all. As the aim was to recruit people from the general public who are interested in the intervention, self-selection and quasi-experiment was the study design that best matched our requirements.

-There is a  detailed program description would enrich the manuscript.

We have added a more detailed explanation of our mobile application, data collection process, and task delivery formats.

It would be interesting to use a flow chart related to participants participation

We agree that flow charts could contribute to the clarity of the drop-out data, but at the same time, it would take a whole page to display 16 dropout points (days). That’s why we decided not to include a chart for dropout data.

  1. c) Results: 

-Results are well described

-Probably the results from reliability coefficients would be more systematic presented in a table

We agree. We have formated both Shapiro-Wilks and Cronbach alpha data into APA format tables and added them to the results section. 

  1. d) Discussion:

-Data are well discussed but should be more articulated with the state of art

Thanks, we added something into the Discussion section.

-It could be interesting to explore the association with expressive writing of Pennebaker, it was mentioned but only shortly

Thanks a lot. We added more info about how we used the expressive writing paradigm in our research study.

-I do not agree with this argument, I think it would be very important to have some baseline evaluations of the target population «Therefore, our EFT-SCP mobile app has the potential to improve well-being and decrease pathology, although we did not measure these important psychological constructs because we did not want to overwhelm participants by requiring them to complete so many questionnaires.»

Thanks, the used scales are validated in Slovakia, and we have norms for them so we know that the baseline average measurements were within the average scora in Slovak general population (Halamová & Kanovský, 2017 and Vráblová & Halamová, 2022). Changed in the manuscript.

-However it would be interesting to discuss some implications of the program nature (techniques and theoretical assumptions), comparisons between the different approaches used in an app and face to face

Thanks, the main difference between mobile app interventions and face-to-face interventions was described in the theoretical background. There are some advantages and disadvantages on both sides and these were described in the -theoretical background (lines 47-61).

The main difference when creating an application for mental health are its specific constraints. Some things are not possible to program into the application (human interaction) but are designed in form of a question->answer via video, audio, and text implementation. Our results suggest that mobile applications even with their constraints are sufficient in changing some aspects of mental health such as self-compassion, self-criticism, and self-protection. In comparison with face-to-face interventions, they are accessible, inexpensive, available to a broad population, and deal with stigma and those are their main advantages over face-to-face interventions. We added it into the Discussion section.

-The relevance for clinical practice is well established

Thank you.